# Theory of highly efficient multiexciton generation in type-II nanorods

Hagai Eshet[1,2], Roi Baer[3], Daniel Neuhauser[4] & Eran Rabani[2,5]

Multiexciton generation, by which more than a single electron–hole pair is generated on optical excitation, is a promising paradigm for pushing the efficiency of solar cells beyond the Shockley–Queisser limit of 31%. Utilizing this paradigm, however, requires the onset energy of multiexciton generation to be close to twice the band gap energy and the efficiency to increase rapidly above this onset. This challenge remains unattainable even using confined nanocrystals, nanorods or nanowires. Here, we show how both goals can be achieved in a nanorod heterostructure with type-II band offsets. Using pseudopotential atomistic calculation on a model type-II semiconductor heterostructure we predict the optimal conditions for controlling multiexciton generation efficiencies at twice the band gap energy. For a finite band offset, this requires a sharp interface along with a reduction of the exciton cooling and may enable a route for breaking the Shockley–Queisser limit.

[1] School of Chemistry, The Sackler Faculty of Exact Sciences, Tel Aviv University,Tel Aviv 69978, Israel. [2] The Raymond and Beverly Sackler Center for Computational Molecular and Materials Science, Tel Aviv University, Tel Aviv 69978, Israel. [3] Fritz Haber Center for Molecular Dynamics, Institute of Chemistry, The Hebrew University of Jerusalem, Jerusalem 91904, Israel. [4] Department of Chemistry, University of California at Los Angeles, Los Angeles, California 90095 USA. [5] Department of Chemistry, University of California and Materials Science Division, Lawrence Berkeley National Laboratory, Berkeley, California 94720, USA. Correspondence and requests for materials should be addressed to R.B. (email: roi.baer@huji.ac.il) or to D.N. (email: dxn@chem.ucla.edu) or to E.R. (email: eran.rabani@berkeley.edu).

ncreasing the photovoltaic power conversion efficiencies in solar cells beyond the Shockley–Queisser (SQ) limit[1] of 31% remains one of the grand challenges of the field. Several theoretical concepts to overcoming this limit, to as much as 66%, have yet to be realized in state-of-the-art photovoltaic devices[2]. One promising direction for collecting hot carriers is based on multiexciton generation (MEG)[3–5] or singlet fission[6], in which more than one electron–hole pair is generated on optical excitation, potentially increasing photovoltaic power conversion efficiencies up to 45% (refs 7,8). To date, solar cells using MEG still show poor performance[9], mainly due to the excessively high MEG onset energy ($E_{on}$).

The necessary conditions for MEG to be technologically useful are that the onset energy $E_{on}$ be close to the lower threshold of twice the quasiparticle band gap ($2E_g$) and that beyond this onset the MEG efficiency (that is, the average number of excitons produced by each absorbed photon) increases rapidly with photon energy[7]. To push the onset of MEG to $2E_g$, the rate of generating multiexcitons needs to exceed the rate of the competing non-radiative exciton cooling channel[10]. Achieving this goal is very different from older concepts of collecting hot carriers, since the timescales for separating charge carriers are very long compared with the timescales of carrier multiplication by MEG.

In bulk semiconductors, due to conservation of linear momentum and energy and due to the rapid non-radiative relaxation of excitons (exciton cooling), the onset energy is typically $E_{on} \approx 5E_g$ (ref. 11). On the other hand, confined semiconductor nanocrystals (NCs) offer improved MEG efficiencies[12,13], with an onset energy below $E_{on} \approx 3E_g$ (refs 14–20), rationalized by several theoretical treatments[10,21–32]. Among these, perhaps the most intuitively appealing theoretical picture of MEG in NCs is that of impact ionization, describing the decay of an excited charged quasiparticle to an identically charged trion, composed of a quasiparticle and an exciton[10,21,22,30]. For efficient MEG, the trion formation rate should surpass the combined rate of decay by all other non-radiative exciton cooling channels. Using the impact excitation formalism and a unified Green's function approach to MEG, we have shown that the interplay of the size scaling of the band gap, Coulomb couplings and density of states (DOS) leads to an overall increase in MEG rates and efficiencies with decreasing NC size[31,33]. The role of the relative magnitude of the electron and hole effective masses has also been discussed recently[34].

Several different approaches have been proposed to further reduce the onset energy of MEG. Gabor et al.[35] used a polarized field to decrease $E_{on}$ in carbon nanotubes photodiodes, a result that was rationalized by a classical description based on field-induced acceleration of the charge carriers[36]. Sandberg et al.[37] studied the role of the shape of nanoparticles on the MEG efficiency and argued for a 60% increase in MEG yields in semiconductor nanorods (NRs) compared with spherical NCs. The role of the diameter and length of the NR on the MEG efficiencies was analysed by considering the scaling of the density of trion states (DOTS) and the Coulomb couplings[38]. Both show distinct scaling from spherical NCs leading to a surprising result that the MEG efficiencies were roughly independent of the NR length[38]. This prediction was later verified by experiments[39,40].

Two additional handles can be used to control MEG efficiencies. The first is based on suppressing the competing channel of phonon emission. Experimental work along this line by Klimov et al. has shown that MEG onset energy in type-II core-shell NCs can be reduced by partially blocking the non-radiative decay channel[5]. Theoretically this mechanism has been studied on very small NC sizes[41,42]. The other, which forms the crux of our approach, is based on controlling the DOTS

near $2E_g$, thereby reducing the onset energy to $E_{on} \approx 2E_g$ and markedly increasing MEG efficiencies near the onset. One way we tried recently to achieve a control of the DOTS was by using seeded NRs heterostructures[43]. This did not lead to a significant increase of MEG efficiency near $2E_g$ (ref. 44), but the analysis showed that these efficiencies increase with seed size, contrary to the case of isolated spherical NCs.

In this work, we report on extremely high MEG efficiencies at the onset energy of $E_{on} \approx 2E_g$ in type-II NR heterostructures, which can be prepared using cation exchange reactions[45]. Our work is motivated by the results of MEG in carbon nanotube photodiodes[35,36], but uses the internal field in type-II NRs as opposed to applying an external polarized field. We show that by using the internal field energy along with a suppression of the cooling rates, MEG efficiencies can increase to over 150% at $2E_g$, significantly above any previously reported result for colloidal nanostructures. The work develops ways for increasing solar cell efficiencies beyond the SQ limit.

## Results

**Model for a type-II nanorod heterostrucure**. To illustrate the core idea and to focus on the effect associated with the role of the internal field, we consider a model system of type-II NR where both building blocks are composed of CdSe with an internal field at the interfacial region of width $\beta$ and maximal value $F \approx \Delta/\beta$, as depicted in the left panel of Fig. 1 ($\Delta$ is defined in the caption of Fig. 1). While more realistic models of type-II nanostructures can be handled within the pseudopotential approach[46], the simplified approach we take allows us to delineate the mechanism of generating multiexctions induced by the internal field only, without the complexity of varying the band structure and/or introducing stress and strain across the hetero-junction.

Before we proceed to compute the MEG efficiencies, we first demonstrate that the optical absorption, most importantly the absorption onset energy, does not change with the internal field. We have calculated the absorption cross section using a stochastic approach suitable for thousands of electrons (see 'Methods' section for more details) within the pseudopotential model for CdSe[47], neglecting the electron–hole interactions which are not important for establishing the dependence of the absorption onset energy on the internal field. In the right panel of Fig. 1 we plot the results for a series of NRs with different internal fields. As can be seen clearly, the absorption onset is independent of the internal field strength within the noise level of the stochastic approach, consistent with the small overlap of the hole and electron wave functions when they are localized on opposite sides of the NR. Moreover, the entire line-shape is very similar for different values of $\Delta$, implying that indeed the band structure of the parent material is preserved with the perturbing internal field, even for rather large potential drops.

**MEG efficiencies in a type-II nanorod heterostrucure**. We now proceed to compute the MEG efficiencies. This is a complicated many body problem involving singly and doubly excited states and their decay via electron–phonon couplings. Progress can only be made with well-defined, controlled approximations. We have recently shown that the direct[24,27,48] or indirect absorption approach[15], the coherent reduced density matrix formalism[23,26] and the impact excitation[10,21,22] all emerge as different limits of a Green's function formalism[33]. The impact excitation model proved to be accurate in comparison with a more elaborate Green's function approach while providing a simple physical picture in which MEG efficiencies become notable if their rates are larger than the competing exciton cooling rates. Furthermore, the MEG process can be described to lowest order in the

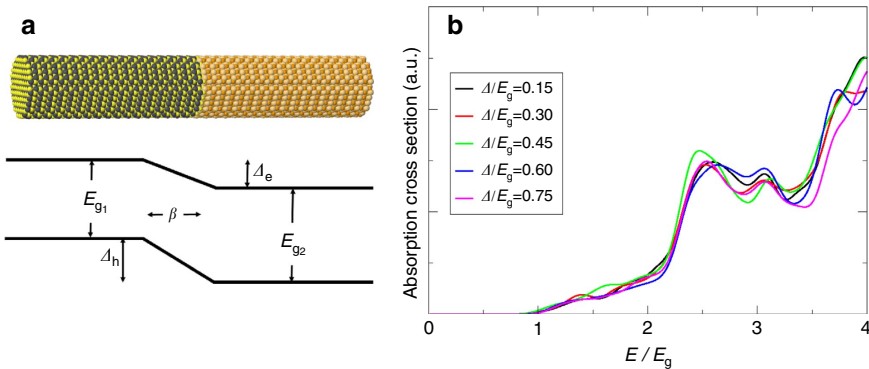

**Figure 1 | A sketch of the type-II nanorod heterostructure and its absorption cross section.** (**a**) A sketch of the type-II NR (sizes of 15 nm length and 3.8 nm diameter, containing $N_e \approx 30{,}000$ electrons) and valence band maximum and conduction band minimum energies along the rod axes. In the calculations reported in this work, we use $E_{g_1} = E_{g_2} \equiv E_g$, $\Delta_h = \Delta_e \equiv \Delta$ and $\beta$ is the range of potential drop, simulated as $V(z) = \Delta(\frac{1}{e^{\beta z}+1} - 1)$ where the center of the NR is at $z = 0$. In realistic materials $\Delta$ is typically on the order of $\sim 0.3$ eV and $E_g$ can range between 0.6 to 2 eV. We use a pseudopotential model developed by Rabani et al.[47] which was fitted to reproduce the band structure of bulk CdSe. (**b**) Absorption cross section computed using a stochastic approach ignoring the electron–hole interactions for different $\Delta$'s for $\beta = 0$. The photon energy $E$ is scaled by the bare CdSe gap $E_g = 2.04$ eV and the concepts discuss carry to materials with lower gaps as well.

Coulomb coupling as a decay of an electron/hole to a negative/positive trion state (composed of two electrons and one hole or vice versa), while the other carrier (hole/electron) is a spectator. For these reasons we adopt the impact excitation approach and implement a stochastic method developed in refs 31,44 to overcome the computational complexity of computing MEG rates in systems with thousands of electrons.

For completeness, we briefly review the approach and the key elements in describing the MEG efficiency $n_{ex}(\omega)$, the number of excitons per absorbed photon. This can be written as an average over sampled electron–hole pair excitations, $S$:

$$n_{ex}(\omega) = \frac{1}{N_S} \sum_S \mathcal{P}_S(\omega) \frac{2\Gamma_S + \gamma}{\Gamma_S + \gamma} \qquad (1)$$

where $N_S$ is the number of sampled excitons (typically $N_S = 200$ is sufficient to converge the results), $E_S = \hbar\omega$ is the photon and exciton energy, $\gamma$ is the exciton cooling rate taken to be energy independent, $\Gamma_S$ the MEG rate for exciton $S$, and $\mathcal{P}_S(\omega) = \rho(\varepsilon_e)\rho(\varepsilon_h)P_S(\omega)$ (where $\rho(\varepsilon)$ is the density of states at energy $\varepsilon$),

$$P_S(\omega) = \frac{|\mu_{0S}|^2 \delta(E_S - \hbar\omega)}{\sum_{S'} |\mu_{0S'}|^2 \delta(E_{S'} - \hbar\omega)} \qquad (2)$$

is the probability of generating exciton $S$ given that a photon of energy $\hbar\omega$ was absorbed. In the above, $\mu_{0S}$ is the transition dipole from the ground state $|0\rangle$ to an excitonic state $|S\rangle$. Finally, the MEG rates, $\Gamma_S$, are given in terms of a sum of negative ($\Gamma_e^-$) and positive ($\Gamma_h^+$) trion formation rates $\Gamma_S = \Gamma_e^- + \Gamma_h^+$, where

$$\Gamma_e^- = \frac{2\pi}{\hbar} \langle W_e^2 \rangle \rho_T^-(\varepsilon_e)$$
$$\Gamma_h^+ = \frac{2\pi}{\hbar} \langle W_h^2 \rangle \rho_T^+(\varepsilon_h) \qquad (3)$$

and $\sqrt{\langle W_{e,h}^2 \rangle}$ and $\rho_T^{-,+}(\varepsilon_{e,h})$ are the average screened Coulomb coupling and the density of trion states, respectively. All the required quantities can be obtained within the pseudopotential approach using the stochastic MEG method[31,44].

In Fig. 2 we plot the calculated MEG efficiencies for a series of NRs with varying internal fields (left panels) and varying exciton cooling rates (right panels) as a function of the photon energy $E = \hbar\omega$. Consistent with above discussion of the onset of absorption, we scale the photon energy of all NRs with the band gap of the unbiased system, $E_g = 2.04$ eV. The left panel of Fig. 2 illustrates an important result, where the onset energy for MEG,

$E_{on}$, decreases markedly with the bias potential. Furthermore, comparing the results of the upper and lower left panels of the figure it is clear that a much larger effect is obtained when the length scale for the bias drop is small, namely for a large field. This latter result is consistent with theoretical predictions of the suppression of Auger processes in core-shell nanocrystals, where a sharp interface between the core and shell leads to faster Auger recombination[49].

Above a certain threshold value of $\Delta$, the onset energy of MEG can be smaller than $2E_g$. This results requires an explanation since the minimal double-exciton energy is twice the energy of the gap, $E_g$. To explain this seemingly unrealistic result, we sketch two possible relaxation mechanisms for generating multiexcitons in the upper panel of Fig. 3. Consider an electron at high energy delocalized along the entire NR. The electron can either generate a negative trion where all carriers are localized in one building block, or can generate a negative, charge separated, trion. On the basis of energy conservation considerations, in the former the theoretical onset of MEG is $E_g - \Delta$ while in the latter it is $E_g - 2\Delta$. Thus, due to the potential drop the onset can be pushed below $2E_g$. We note that for the charge separated negative trion, the hole orbital must overlap one of the final electron orbitals. The lower panel of Fig. 3 illustrates that this can indeed happen. However, this overlap is rather small and the central contribution comes from the localized trion state. We conclude that the dominant mechanism will push the theoretical onset energy to $2E_g - \Delta$.

**Analysis of the multiexciton generation efficiencies.** While a finite value of $\Delta$ is required to separate the charge carriers in type-II solar cell junctions, the value of $\Delta$ used to increase the MEG efficiency as depicted in the left panels of Fig. 2, will lead to a significant decrease in the photovoltage and as a result, to an overall lower solar cell performance[50]. As shown in the right panels of Fig. 2, this issue is mitigated when the exciton cooling lifetime is increased above 10 ps (which, in fact, can be induced by an pronounced electron–hole charge separation[5]), leading to a MEG onset energy below $2E_g$, even at moderate values of $\Delta$. It is clear, however, that only by combining a non-negligible internal field and a mechanism to slow down the cooling of excitons by phonon emission, the MEG efficiency can be controlled at the desired onset, while either one of these mechanisms separately will not be sufficient.

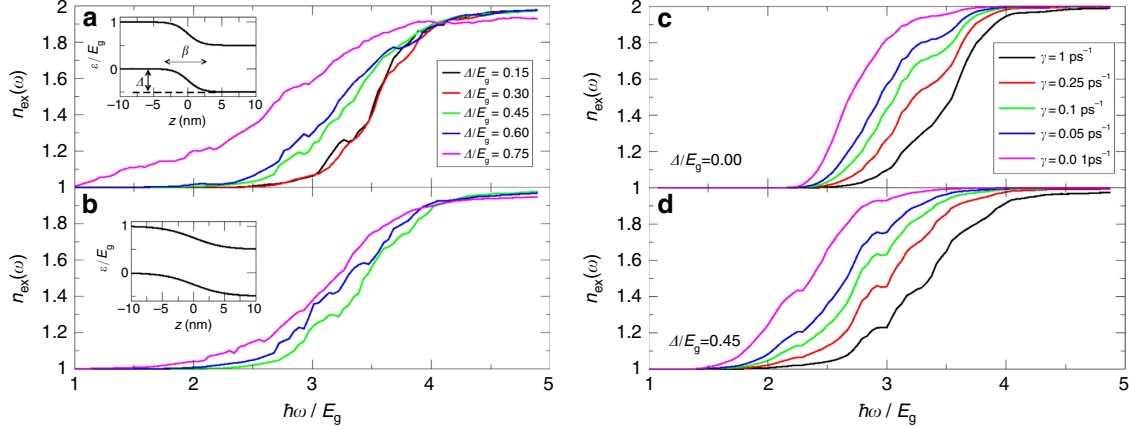

**Figure 2 | MEG efficiencies for a type-II nanorod heterostructure.** (**a,b**): Plots of the biexciton generation efficiency as a function of energy for several different value of $\Delta$. The cooling rate is $\gamma = 1\,\text{ps}^{-1}$ and the range parameter is $\beta = 2.5\,\text{nm}$ for (**a**) and $\beta = 1\,\text{nm}$ for (**b**). Note that the results for the two lower values of $\Delta$ for $\beta = 2.5\,\text{nm}$ are similar to those of $\Delta = 0.45E_g$, and thus are not shown. Insets show the valance band maximum and conduction band minimum energies along the rod axes. (**c,d**): Plots of the biexciton generation efficiency as a function of energy for different values of $\gamma$ (the exciton cooling rate) for $\Delta = 0E_g$ (**c**) and $\Delta = 0.45E_g$ (**d**).

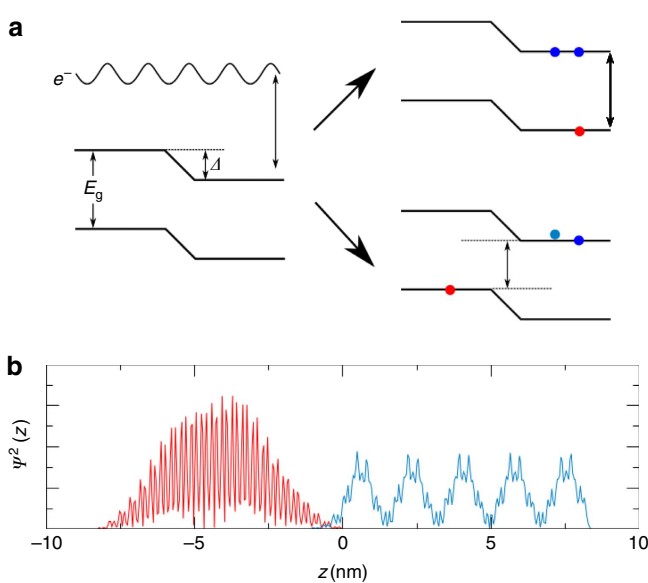

**Figure 3 | MEG mechanism in type-II nanorod heterostructure.** (**a**) A sketch of the paths (labelled by arrows) for forming a negative trion by an excited electron delocalize along the nanorod. The excited electron can decay to a negative trion where all charge carrier are localized on one side of the nanorod (upper path) or to a charge seprated trion (lower path). (**b**) A typical hole (red) and electron (blue) projected densities along the z axis showing overlapping electron–hole densities required for a non-vanishing probability to form a charge separated trion. The results are for $\Delta/E_g = 0.75$ and $\beta = 1\,\text{nm}$.

The increase of MEG rate near $2E_g$ is further analysed in Fig. 4, where we plot the DOTS (upper panel) and the average Coulomb coupling for electrons (lower panel). The density of negative and positive trion states ($\rho_T^-(\varepsilon_e)$ and $\rho_T^+(\varepsilon_h)$, respectively) is plotted along the positive and negative energies, respectively. Each is measured from the bottom/top of the conduction/valence bands of the unbiased CdSe NR. The DOTS is obtained by a triple convolution of the DOS not shown here. For the negative DOTS, the triple convolution involves the DOS of two electrons (that is, the DOS above the Fermi energy) and one hole (that is, the DOS below the Fermi energy) while for the positive DOTS it involves the triple convolution of the DOS of two holes and one electron. Common to both DOTS is the increase in $\rho_T^\pm(\varepsilon_e)$ at $\varepsilon = \pm E_g$ above/below the conduction/valance band minimum/maximum. Note that we use a semi-logarithmic scale and thus, the magnitude of the effect is rather dramatic with increasing bias. The positive DOTS is typically higher than the negative DOTS at the energies measured from the corresponding band edge, a result that can be explained by the larger DOS of holes[44]. Furthermore, the negative DOTS shows a stronger dependence on the internal bias potential.

The increase in the DOTS at energies $\pm E_g$ above and below the corresponding band extremes is a necessary but not sufficient condition for pushing the onset of MEG to $2E_g$. In addition, the coupling of the excited electron–hole pair need not vanish. While the average Coulomb coupling for electrons shown in the lower panel of Fig. 4 is nearly independent of energy at high energies, consistent with the picture emerging for NCs (ref. 31) and NRs (ref. 38), in the important range of $\varepsilon \approx E_g$ there is a notable small increase in $\sqrt{\langle W_e^2 \rangle}$ as the internal field is increased due to a change in the character of the trion states to which the electron (hole) is coupled. The increase in the DOTS and in small increase in $\sqrt{\langle W_e^2 \rangle}$ leads to a significant soaring of the trion formation rate above the exciton cooling rate, $\gamma$. This interplay of DOSs and Coulomb coupling and the energy dependence is reflected in an increase of MEG efficiencies and a decrease of $E_{on}$, as discussed above.

In summary, using MEG in nanostructures for breaking the SQ limit requires a significant increase in the MEG efficiencies at $2E_g$. So far, this has been elusive and neither spherical NCs nor core-shell NCs nor NRs push the onset of MEG to $2E_g$. Here, we provide the theoretical foundation for the design of materials with large MEG efficiencies near $2E_g$. Our predictions are based on an atomistic model for MEG within a fully quantum mechanical description, which has been shown to be accurate in postdicting and predicting MEG in NCs and NRs. The two key elements for pushing down the MEG onset energy are: (a) Using the internal field ($\Delta/\beta$) in type-II heterostructures with a rather sharp interface between the two building blocks and (b) slowing down the exciton cooling rate by reducing the electron–phonon

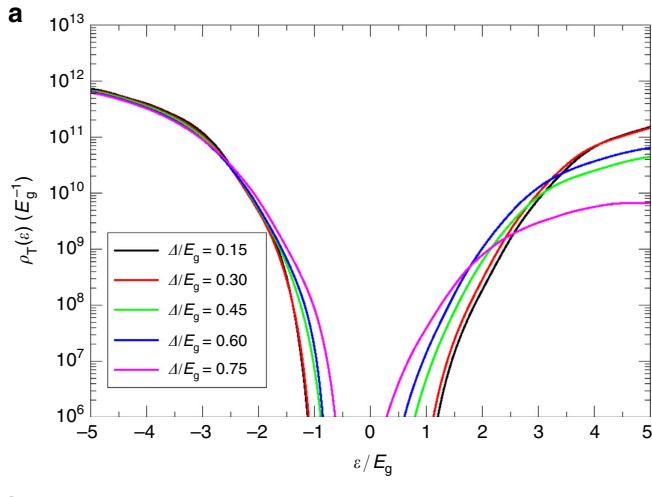

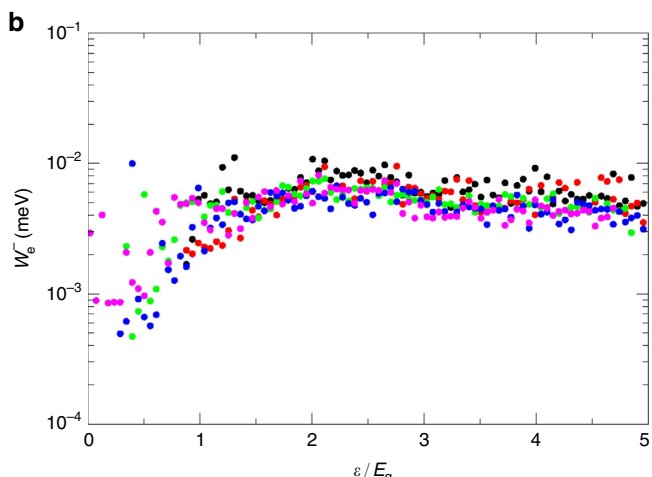

**Figure 4 | Density of trion states and effective Coulomb couplings in a type-II NR heterostructure.** DOTS (**a**) and average Coulomb couplings $W_e^- \equiv \sqrt{\langle W_e^2 \rangle} = \sqrt{\frac{\Gamma_e^-(\varepsilon)}{2\pi\rho_T^-(\varepsilon)}}$ (**b**) for different $\Delta$'s for $\beta = 1$ nm. The positive (hole) DOTS are plotted on negative energies and the negative (electron) DOTS are plotted on positive energies.

couplings. Only the two combined effects will lead to a significant increase in the MEG efficiencies at $2E_g$. We find that the MEG efficiency at the onset of $2E_g$ increases with increasing band offsets ($\Delta$), increases with decreasing values of $\beta$ (that is, sharper interface between the two building block materials) and increases with slower cooling rates, $\gamma$. For the current CdSe NR system and for realistic parameters, the MEG efficiency at $2E_g$ can be as large as $10\% - 20\%$. This should be compared with an onset of $3E_g$ and vanishing carrier multiplications at $2E_g$ of previously studied colloidal nanostructures.

Further development is required to utilize the current scenario in type-II single cell junctions. For an optimal performance, one requires the design of materials with gaps close to 0.7 eV (ref. 7), a sharp interface between the building blocks and a moderate value of the band offset, $\Delta$, not to reduce significantly the photocurrent voltage. Moreover, an efficient and rapid scheme to collect multiple charge carriers is requires before Auger recombination and/or other relaxation process step in. While some of these targets are already achievable by colloidal chemistry synthesis, the reduction of the cooling rate and the rapid collection of multiple charges requires the development of novel schemes along the lines recently proposed for core/shell nanocrystals[5].

## Methods

**Computational method.** All calculations were preformed within the pseudopotential model for CdSe (ref. 47) implemented in a real-space grid of size $96 \times 96 \times 384$ and spacing of 0.8 a.u. sufficient to converge the results. The NR included 3,600 Cd and 3,600 Se atoms and ligand potentials were used to passivate the dangling bonds[47]. The description of the electronic properties of this system is beyond the current capabilities of deterministic approaches. Therefore, we resort to our recently developed stochastic formalism[31]. For the MEG efficiency given by equation (1) we have carried the following steps:

First, the DOSs, $\rho(\varepsilon)$, were obtained using a stochastic trace formula[31] with an interpolation polynomial of length $N_C = 16,384$ averaged over 50 stochastic iterations. The negative and positive trion DOSs $\left(\rho_T^-(\varepsilon)/\rho_T^+(\varepsilon)\right)$ were then generated by a triple convolution of the single-particle DOSs. More details are given in ref. 31. Next, negative and positive trion formation rates in equation (3) were generated using a stochastic procedure[31]. This involves filtering electron and hole states at a target energy and filtering a resonant negative and positive trion state, respectively, using an interpolation polynomial of length $N_C = 16,384$. For each interpolation polynomial filter we have generated 16 electron states and 16 hole states and for each particle a corresponding resonant trion state. The rates were averaged over 10,000 stochastic realizations. In addition, for each electron or hole obtained in the previous step, we have filtered several (10) hole or electron states, respectively, and for each electron–hole pair (20 independent pairs), we calculated the transition dipole $\mu_{0S}$ at energy $E_S$ and obtained $\mathcal{P}_S(\omega)$ given by equation (2). Since we only sample electron–hole pairs, instead of calculating them all, we have multiplied the result by the density of electrons and holes $\rho(\varepsilon_e)\rho(\varepsilon_h)$ at the reference energy of the electron ($\varepsilon_e$) and hole ($\varepsilon_h$). For more information see ref. 44. Finally, we have used $\Gamma_e^-$ and $\Gamma_h^+$ to generate the MEG rate for each exciton: $\Gamma_S = \Gamma_e^- + \Gamma_h^+$ and then obtain $n_{ex}(\omega)$ by performing the average in equation (1) for $N_S = 50,000$.

The absorption cross section, $\sigma(\omega) \propto \omega r(\omega)$ is given in terms of the golden rule absorption rate, $r(\omega) = \frac{\mathscr{E}^2}{\hbar} \sum_S \int_{-\infty}^{\infty} dt\, e^{i\omega t} \langle S|\mu|0\rangle\langle 0(t)|\mu|S(t)\rangle$, where $\mu$ is the dipole operator, $\mathscr{E}$ is the amplitude of the electromagnetic field, $|0\rangle$ is the ground state and $|S\rangle$ is an electron–hole pair state. Instead of summing overall electron–hole pairs, an impractical task for the studied NR size, we used the following stochastic formula: $r(\omega) = \frac{\mathscr{E}^2}{\hbar N_\zeta} \sum_\zeta \int_{-\infty}^{\infty} dt\, e^{i\omega t} \langle \zeta_e|\mu|\zeta_h\rangle\langle \zeta_h(t)|\mu|\zeta_e(t)\rangle$, where $\zeta_h$ is a projected random orbital on the occupied space and $\zeta_e$ is a project random orbital on the unoccupied space (see more details about random stochastic orbitals in ref. 51). We used $N_\zeta = 10,000$ to average the rate of absorption and $N_t = 10^5$ propagation steps with a time step of 0.025 a.u. to propagate the electron–hole stochastic orbitals.

**Data availability.** The data that support the findings of this study are available from the corresponding authors on request.

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

## Acknowledgements

R.B. gratefully acknowledges support for his sabbatical visit by the Pitzer Center and the Kavli Institute of the University of California, Berkeley and the ISF grant 189/14. E.R. and D.N. acknowledge support from NSF grants CHE-1465064 and CHE-1112500, respectively.

## Authors contributions

H.E. performed the calculations and analysed the data, R.B., D.N. and E.R. developed the theoretical framework and analysed the data, all authors co-wrote the paper.

## Additional information

**Competing financial interests:** The authors declare no competing financial interests.

