## [Peer Review File · Nature Communications]

Reviewer #1 (Remarks to the Author)

A computational investigation is presented that explores the use of indirect excitons at type II heterojunctions to increase the efficiency of multiple exciton generation. This is novel idea with potentially significant technological applications. The premise is clearly laid out and well-supported by a well-established computational methodology. There are several issues that need to be addressed however, as listed below. Perhaps most importantly, though, the authors need to go beyond a parametric study of the key parameters. They have focused on a particular material system for which reasonable estimates for these parameters exist. With such parameters in hand, it should be possible to estimate what sort of impact such type II junction may have on thermodynamic efficiency-i.e. a tie needs to be made back to the original focus on the S-Q limit. If this turns out to be significant, then I recommend publication.

[1] Why is it that the absorption spectrum is not influenced by the heterojunction? In other computational studies of admittedly different systems-DOI: 10.1039/c4cp03042g-this is not the case. The authors need to explain why absorption is unaffected while trion formation is.

[2] Figure 2 (left) shows that the efficiency is strongly influenced by the value of beta (interface width). The authors choose 2.5 and 1.0 nm to demonstrate this. What is a physically reasonable value for beta though? This is something that should be known for the system being studied.

[3] Figure 2 (middle) shows that the exciton cooling rate has a strong influence on the efficiency. In fact, for cooling rates faster than 0.05/ps the effect sought disappears. Why is this? Equally important, the authors need to justify the exciton cooling rates used. What is a reasonable exciton cooling rate for such materials. This should be known. As an aside, it would be helpful to define the units of this cooling rate-e.g. is it the average rate of hopping between energy bands, between the lowest two bands, etc. This figure should be broken up into 2 figures, with the model at right a separate figure. I might have missed it, but I cannot find reference to the lower right panel of this figure.

[4] Figure 3 (left) explores how the effect of band offset (Δ) on the density of trion states. Consistent with my comments above, the authors need to provide estimates for this band offset. It must be known for the material system being studied.

[5] Figure 3 (right) would benefit from curves that represent the mean values of each set of dots. Shading around the key focus on ratios near 1 would help to guide the eye of readers.

[6] The first paragraph at the bottom right of page 3 is, if I understand the work correctly, the key result from which the increase in efficiency derives. It would be useful to flesh this idea out earlier in the paper so that the reader understands how the efficiency gain is achieved prior to working through the analysis.

[7] The conclusions need to include an estimate for the efficiency of this particular system given established values or best estimates for the key parameters (Δ , β , γ). This can be followed up by a brief note suggesting other systems for which a modification of such values would lead to improved efficiencies-i.e. how far can this be exploited? The authors have discovered something very novel here, but there is no effort to give the reader a quantitative sense for how important it is.

[8] It would be helpful to have an estimate for the thermodynamic efficiency of a particular CdSe nanorod. If this is not possible, the authors should explain why.

[9] How important is the particular interface geometry? The authors should discuss whether or not such efficiency gains might be possible with core-shell quantum dots and core-shell rods. Along the same lines, is it possible to have spatially separated quantum cutting that benefits in the same

way from indirect excitons?

[11] The first sentence of the abstract is a shortened version of a sentence from the introduction, but the last portion has been cut so much that the meaning is confused. It is not apparent that this part of the sentence is intended to convey that the "average number of excitons produced by each absorbed photon increases rapidly with photon energy".

[10] There are a number of small grammar and spelling errors that can easily be fixed:

--end of paragraph at top/left page 2: "isolate" -> "isolated"

--bottom paragraph at left on page 2: Define Delta. Also, "F" seems to never be used other than at this one point. Is it really necessary?

--middle paragraph at right on page 2: "As clearly can be seen," -> "As can be clearly seen"

--bottom paragraph at right on page 2: "well-define" -> "well-defined"

--caption of Figure 2: "biexciton" is spelled wrong twice

--middle paragraph at right on page 3: "illustrate" -> "illustrates"

--top left paragraph on page 4: comma has space before it

[11] The figures tend to appear before they are discussed in the text. This is confusing and needs to be corrected.

Reviewer #2 (Remarks to the Author)

This paper represents a theoretical study of multiexciton generation (MEG) in model type-II hetero-nanorods (NRs) with both parts comprised of CdSe-like material with identical band gaps (E_g) but an arbitrary band offset (Delta) which is one of the parameters of the calculations. The authors calculate linear absorption spectra of these structures and show that they are virtually independent on Delta. On the other hand, the calculated MEG yields show a considerable increase with increasing Delta, which is interpreted in terms of the effect of increasing electric field. I find these results of potential interest. However, I have several questions regarding their analysis and interpretation.

1. The authors consider that the band gap of their structures is the same as that of CdSe (E_g). However, for type-II hetero-nanocrystals, the actual band gap is defined by the difference in electron and hole energies at the hetero-junction, i.e., $E_{g1,2} = E_g - \Delta$. This energy defines the emission wavelength of hetero-structures as well as the onset of optical absorption. This leads me to the first question: Absorption spectrum in Fig. 1b has a tail extending to 1 eV. It's likely due to the indirect transition at the interface. Why the onset of indirect absorption does not change with Delta? It must be directly linked to $E_{g1,2} = E_g - \Delta$.

2. I have a problem with interpretation of the data in terms of electric field. The potential in Fig. 1a indeed resembles the structure of a p-n junction with the internal field within the depletion layer. However, the structures considered in this paper seem to be intrinsic, i.e., do not contain dopants. Hence, there is no field in the structures. Does the Hamiltonian used in the calculations contain the field term? - Probably not. Hence, the field in system is zero. The interpretation in terms of electric field might confuse readers.

3. My impression is that the main reason for observed enhancement is a trivial effect of reduction of the effective band gap ($E_{g1,2} = E_g - \Delta$) when Delta is increased. How do the data in Fig. 2

(left panel) look if they are re-plotted as a function of $E_{g1,2}$? The authors do need to discuss the distinction between the effect of a trivial reduction of the band-gap and some other effects, which might take place in these type-II NRs.

4. The authors briefly discuss the effect of Δ on photovoltage. In fact, I think they have a misprint on p. 4 where they state that the proposed scheme will increase voltage. The proposed scheme actually should decrease the photovoltage in proportion to the introduced band offset Δ . The extracted voltage will be "pinned" by the indirect band gap $E_{g1,2}$. The question is what is the net gain from the proposed scheme in terms of the power output? The authors should discuss this important aspect of their work in the manuscript.

5. I am not sure that I understand authors' reasoning regarding slowed cooling and its effect on the overall efficiency. They focus on calculations of the effect near the MEG threshold when following impact ionization the carriers reside in band-edge states. They are already "cooled." Do the authors imply that their scheme is based on both MEG and hot-electron-cell ideas? If so, this needs to be discussed in much greater detail when it is discussed in the present version of the manuscript.

To summarize, this work is potentially useful. However, the authors need to provide strong evidence that the results they report are not derived solely from the effective reduction of the band gap in type-II nanostructures. They should also clarify the concept of "electric field" used in their model. At present their interpretation is quite confusing as the structures are undoped. They should also re-analyze their results in terms of an indirect band gap ($E_{g1,2}$) but not only the unperturbed band gap (E_g). In the present form, this paper is not ready for publication.

Reviewer #3 (Remarks to the Author)

Please find the review report on the manuscript entitled "Highly efficient multiexciton generation in type II nanorods (NCOMMS-16-11357)" below:

What are the major claims of the paper?

In the present manuscript, Prof Rabani and his co-workers have provided a theoretical treatment of the pathways to increase the efficiencies of multiple exciton generation (MEG) in quantum nano systems. The authors have considered CdSe nanorod as a model system and found that the MEG efficiency can be increased by a significant amount beyond Shockley-Queisser limit, by considering the joint effects of the internal field and slowing down the exciton cooling rate. They have also rationalized type-II heterostructure as the primary requirement for increasing the efficiency of MEG. The role of the finite band offset in type-II heterostructures to separate the charge carriers and to reduce the electron-phonon couplings for increasing the MEG efficiency has been described in detail. Finally, the authors have shown the variation of both the coulomb couplings as well as the trion density of states as a function of the onset energy to show their effects on MEG efficiency.

Are they novel and will they be of interest to others in this field?

The present manuscript deals with a very timely problem in the field of solar cells where increasing the efficiency of the solar cell performance beyond the Shockley-Queisser limit (31%) is one of the most challenging problems to the entire scientific community involved. This work presents a unique idea for increasing the efficiency of MEG in quantum nano systems. The authors provide valuable information regarding the hybrid type-II nanorod and how the MEG efficiencies are related to the internal field, coulomb coupling and density of the trion states. This work will certainly open up a new direction and would be very much interesting to both experimental and theoretical researchers.

If the conclusions are not original, it would be helpful if you could provide relevant references
The conclusions are original.

Is the work convincing, and if not, what further evidence would be required to strengthen the conclusions?

The results are convincing and authors have provided sufficient evidences for drawing the conclusions. The theoretical methods developed by Rabani et al. are already well established. But, it will be more beneficial to the general readers if authors discuss a little bit about the critical limit of the parameters (such as internal field etc.) relating to the efficiencies of MEG near $2E_g$. Also it will be very interesting (if possible) to emphasize the trend of MEG efficiencies by varying the size of the nanorod studied.

On a more subjective note, do you feel that the paper will influence thinking in the field?

Rabani and his co-workers have been successfully able to show that the MEG efficiency can be increased by a significant amount near the $2E_g$ onset energy by considering the joint effect of the internal field and slowing down the exciton cooling rate. The conclusions made from this work will definitely influence research pertaining to solar cells and their applications.

Is the manuscript clearly written? If not, how could it be made more accessible?

Yes, the manuscript is written well.

Could the manuscript be shortened to aid communication of the most important findings?

It can be published in the present format.

Have the authors done themselves justice without overselling their claims?

Yes.

Have they been fair in their treatment of previous literature?

Yes.

Have they provided sufficient methodological detail that the experiments could be reproduced?

This is a theoretical work, the methods used are well established.

Is the statistical analysis of the data sound?

Yes.

Should the authors be asked to provide further data or methodological information to help others replicate their work? (Such data might include source code for modeling studies, detailed protocols or mathematical derivations).

Not necessary.

Are there any special ethical concerns arising from the use of animals or human subjects?

No.

Suggestions for Modifications:

- In figure 2, left panel, both the upper and lower results show the biexciton generation efficiency as a function of energy for different values of Δ and for two different values of β . It will be nice if the number of curves were same for both the cases. In addition, the whole figure needs better representation and clarity.
- In figure 3, left panel, density of both positive and negative trion states are shown with different Δ values. The negative trion densities for various Δ values follow almost the same trend, whereas positive trion density states vary with different Δ values. Please provide explanation of reasons behind this. Also, the value of density is higher for negative trion than that of the positive one in particular around the onset energy (E_{on}). Please make a brief discussion on this.

In conclusion, I recommend that the paper be published after minor revision.

Referee 1

A computational investigation is presented that explores the use of indirect excitons at type II heterojunctions to increase the efficiency of multiple exciton generation. This is novel idea with potentially significant technological applications. The premise is clearly laid out and well-supported by a well-established computational methodology. There are several issues that need to be addressed however, as listed below. Perhaps most importantly, though, the authors need to go beyond a parametric study of the key parameters. They have focused on a particular material system for which reasonable estimates for these parameters exist. With such parameters in hand, it should be possible to estimate what sort of impact such type II junction may have on thermodynamic efficiency-i.e. a tie needs to be made back to the original focus on the S-Q limit. If this turns out to be significant, then I recommend publication.

We thank the referee for his/her comments and support of our findings. We would like to note that the manuscript is mainly concerned with a basic-research discovery and not with an actual application, and thus the focus of the work was on the former. It is meant for a guide for the experimentalists to develop materials with tailored properties for improved MEG. Regarding the thermodynamic efficiency, we address this in point 8 below.

Specific points of referee 1:

1. *Why is it that the absorption spectrum is not influenced by the heterojunction? In other computational studies of admittedly different systems-DOI: 10.1039/c4cp03042g-this is not the case. The authors need to explain why absorption is unaffected while trion formation is.*

First, we would like to note that the purpose of the two studies is very different. We deliberately consider a system which keeps the original absorption by design, while Lusk and coworkers have the opposite goal, looking for a handle to control the spectrum. The fact that the absorption onset is not affected in the type-II structure is a consequence of the negligible overlap of the **lowest** electron and **highest** hole states on either side of the junction. Namely, the orbitals near the conduction band minimum on the left do not overlap the orbitals near the valance band maximum on the right. For trion formation, there are higher energy orbitals involved which span the entire nanorod, rather than being localized on one side. In the elegant study reported by Lusk and coworkers this is not the case and the ligands' orbitals overlap significantly with the electron/hole orbitals of the silicon core.

2. *Figure 2 (left) shows that the efficiency is strongly influenced by the value of beta (interface width). The authors choose 2.5 and 1.0 nm to demonstrate this. What is a physically reasonable value for beta though? This is something that should be known for the system being studied.*

This is an excellent point. In heterojunction nanorods prepared by cation exchange the belief is that the interface does not span more than two layers, which is on the order of 1nm. However, the length scale associated with the interface depends on the method of preparation. Our work illustrates that MEG becomes more efficient for shorter length scales. Utilizing the principles described in our work would require synthetic methods that result in a sharp interface.

3. *Figure 2 (middle) shows that the exciton cooling rate has a strong influence on the efficiency. In fact,*

for cooling rates faster than 0.05/ps the effect sought disappears. Why is this? Equally important, the authors need to justify the exciton cooling rates used. What is a reasonable exciton cooling rate for such materials. This should be known. As an aside, it would be helpful to define the units of this cooling rate-e.g. is it the average rate of hopping between energy bands, between the lowest two bands, etc. This figure should be broken up into 2 figures, with the model at right a separate figure. I might have missed it, but I cannot find reference to the lower right panel of this figure.

Regarding the first part of the referee’s question, we assume that the referee is asking about the effect of increasing the efficiency at $2E_g$. For a high MEG efficiency at $2E_g$, the MEG rate needs to be faster than the cooling rate. For the choice of bias and parameters shown in the lower middle panel of Fig. 2 (now right panel of Fig. 2), the MEG rate at $2E_g$ is just above $1/10\text{ps}^{-1}$.

Regarding the second question, in fact, the cooling rate of type-II structures is not known, even near the band edge. In spherical nanocrystals the rate is on the order of a ps. Thus, the slower rates that give rise to a large increase in the MEG efficiency are somewhat slower than those of spherical nanocrystals. However, the crucial point here is that it is not sufficient to control the rates. One also needs to control the internal field. Either one separately will not do the job of increasing the MEG efficiencies.

Regarding the definition of the cooling rate, we first would like to note that we take it to be energy independent. In the revised manuscript on page 3, right column, below Eq. (1) we have added a note to emphasize this: “ γ is the exciton cooling rate, taken to be energy independent,...” In principle, γ is the total cooling rate from an excitonic states (namely the summation of transition to all other excitonic states induced by electron-phonon coupling), but since we assume it is energy independent, it can be related to a specific transition between two excitonic states.

We have followed the referee’s advice and broke Fig. 2 into two separate figures. On page 4, right column, we have added a reference to the new Fig. 3: “The lower right panel of Fig. 3 illustrates that this can indeed happen. However, this overlap is rather small and the central contribution comes from the localized trion state. We conclude that the dominant mechanism will push the theoretical onset energy to $2E_g - \Delta$.”

4. *Figure 3 (left) explores how the effect of band offset (Δ) on the density of trion states. Consistent with my comments above, the authors need to provide estimates for this band offset. It must be known for the material system being studied.*

The band offset is known and depends on the choice of the materials used to construct the heterojunction. It is typically in the range of 0.3 eV. In units of the band gap, this can range between 0.1 in large band gap materials to even 0.5 in small band gap materials. Our study covers a range of band offsets in order to explore the effect of changing the internal field, rather than focusing on a specific choice of materials. We have added a note in main text on page 2, in the caption of Fig. 1, to reflect the referee’s request: “In realistic materials Δ is typically on the order of ~ 0.3 eV and E_g can range between 0.6 and 2 eV”.

5. *Figure 3 (right) would benefit from curves that represent the mean values of each set of dots. Shading around the key focus on ratios near 1 would help to guide the eye of readers.*

It is in fact important to keep sets of dots in Fig. 4 (was Fig. 3) because they demonstrate the crucial message that the DOTS increases with Δ even near $\varepsilon/E \approx 1$ while W_e^- stays rather insensitive to Δ (or slightly increases). Thus, instead of adding a guide to the eye, we have emphasized this conclusion in the discussion of the results of Fig. 4 on page 5, left column, second paragraph: “While the average Coulomb coupling for electrons shown in the lower panel of Fig. 4 is nearly independent of energy at high energies, consistent with the picture emerging for NCs and NRs, in the important range of $\varepsilon \sim E_g$ there is a notable small increase in $\sqrt{\langle W_e^2 \rangle}$ as the internal field is increased due to a change in the character of the trion states to which the electron (hole) is coupled. The increase in the DOTS and the small increase in $\sqrt{\langle W_e^2 \rangle}$ leads to a significant soaring of the trion formation rate above the exciton cooling rate, γ .”

6. *The first paragraph at the bottom right of page 3 is, if I understand the work correctly, the key result from which the increase in efficiency derives. It would be useful to flesh this idea out earlier in the paper so that the reader understands how the efficiency gain is achieved prior to working through the analysis.*

We agree with the referee and thus, added the following sentence in the introduction on page 2, right column, top paragraph: “We show that by using the internal field energy along with a suppression of the cooling rates, MEG efficiencies can increase to over 150% at $2E_g$, significantly above any previously reported result for colloidal nanostructures.”

7. *The conclusions need to include an estimate for the efficiency of this particular system given established values or best estimates for the key parameters (Δ , β , γ). This can be followed up by a brief note suggesting other systems for which a modification of such values would lead to improved efficiencies-i.e. how far can this be exploited? The authors have discovered something very novel here, but there is no effort to give the reader a quantitative sense for how important it is.*

We have added a paragraph at the end of the conclusions on page 5, left column, bottom paragraph: “In summary, using MEG in nanostructures for breaking the SQ limit requires a significant increase in the MEG efficiencies at $2E_g$. So far, this has been elusive and neither spherical NCs nor core-shell NCs nor NRs push the onset of MEG to $2E_g$. Here, we provide the theoretical foundation for the design of materials with large MEG efficiencies near $2E_g$. Our predictions are based on an atomistic model for MEG within a fully quantum mechanical description, which has been shown to be accurate in postdicting and predicting MEG in NCs and NRs. The two key elements for pushing down the MEG onset energy are: (a) Using the internal field (Δ/β) in type-II heterostructures with a rather sharp interface between the two building blocks and (b) slowing down the exciton cooling rate by reducing the electron-phonon couplings. Only the two combined effects will lead to a significant increase in the MEG efficiencies at $2E_g$. We find that the MEG efficiency at the onset of $2E_g$ increases with increasing band offsets (Δ), increases with decreasing values of β (i.e., sharper interface between the two building block materials), and increases with slower cooling rates, γ . For the current CdSe

nanorod system and for realistic parameters, the MEG efficiency at $2E_g$ can be as large as 10 – 20%. This should be compared with an onset of $3E_g$ and vanishing carrier multiplications at $2E_g$ of previously studied colloidal nanostructures.”

8. *It would be helpful to have an estimate for the thermodynamic efficiency of a particular CdSe nanorod. If this is not possible, the authors should explain why.*

Generic estimations the thermodynamic efficiency were published, for example, in figure 3 of reference 7. For completeness, we include a plot of the cell efficiency as a function of the band gap energy in this response. This plots is taken from Ref. 7.

The black curve is the SQ limit and has a maximum of 31% at $\approx 1.3\text{eV}$. The red curve (line 1) assumes maximal MEG efficiency, namely 2 electron-hole pairs are generated at twice the gap, 3 electron hole pairs at 3 times the gap, etc. The maximum efficiency is 45% for a gap of $\approx 0.7\text{eV}$. This is the optimal case. The green curve assumes that the onset is at twice the gap and increases linearly to 3 times the gap. The maximum efficiency is 37% at $\approx 0.95\text{eV}$ (this is close to the situation we observe). The blue curve assume the onset is at 2.5 times the gap and the maximum efficiency does not exceed the SQ limit. Based on these calculations, it is clear that (a) one has to push the onset of MEG to twice the gap and (b) the MEG efficiency should increase rapidly above the onset. Both of these challenges are met in our work.

Obviously, CdSe is not the material of choice for harvesting light. Moreover, based on the above figure, even optimal MEG conditions will not lead to an increase of the thermodynamic efficiency in CdSe. This is because CdSe has a large optical gap (in the current nanorod studied the quantum confined gap is $\approx 2\text{eV}$). So why use CdSe in our calculations? The main reason is that this is the most studied colloidal nanostructured system and thus, our predictions can likely be verified experimentally. Moreover, since it is highly studied, very accurate electronic structure models exist and provide excellent agreement for the quantum confinement energies, excitons binding energies, hyper-fine structure,

multiexciton generation, Auger rates, etc, in comparison to experiments. Other materials, like silicon, InAs, or PbSe are not direct band gaps, or have a larger differences between the electron and hole effective masses, or require the inclusion of spin-orbit interactions which complicates the calculations.

What can we learn from the predictions on CdSe? We have developed basic concepts for moving the onset energy of MEG to the relevant regime of $2 - 2.5E_g$. These concepts are general and are valid for other materials with band gaps that are more suitable for light harvesting applications. In fact, for spherical nanocrystals, we have shown previously that the MEG efficiencies are nearly material independent once scaled by the gap. This observation should also hold for other material geometries, like the type-II nanorod structure studied here. Therefore, our prediction and the developed concepts should serve as a guide for experimental groups to develop materials with highly efficiency MEG and increasing thermodynamics efficiencies beyond the SQ limit.

9. *How important is the particular interface geometry? The authors should discuss whether or not such efficiency gains might be possible with core-shell quantum dots and core-shell rods. Along the same lines, is it possible to have spatially separated quantum cutting that benefits in the same way from indirect excitons?*

Similar effect should be observed in core-shell systems. However, in core-shell systems it is difficult to collect the charge localized at the core since it is surrounded by the shell. This is not the case in type-II structure with the geometry described in our work. Regarding the second question, by “indirect exciton” the reviewer means excitation in an indirect band gap material or excitation in type-II structures from one material to the other? The latter will not lead to an increase in the MEG efficiencies and, as emphasized in our manuscript, the optical spectrum need not change. For the former, we have not performed the calculations of indirect materials with type-II band alignment and thus, we cannot answer this question. However, we note that in spherical silicon nanocrystals, which are weakly indirect (strictly indirect only for the bulk limit), the MEG efficiencies are similar to CdSe when the results are scaled for the optical gap.

10. *The first sentence of the abstract is a shortened version of a sentence from the introduction, but the last portion has been cut so much that the meaning is confused. It is not apparent that this part of the sentence is intended to convey that the “average number of excitons produced by each absorbed photon increases rapidly with photon energy”.*

We have rewritten the first sentence of the abstract on page 1: “Multiexciton generation (MEG) is a promising paradigm for pushing the efficiency of solar cells beyond the Shockley-Queisser (SQ) limit. Utilizing MEG requires that the onset energy be close to twice the active material band gap energy and that MEG efficiency will increase rapidly above this onset.”

11. *There are a number of small grammar and spelling errors that can easily be fixed*

We have fixed all typographical errors.

12. *The figures tend to appear before they are discussed in the text. This is confusing and needs to be*

corrected.

We have moved the figures to appear close to where they are discussed in the text.

Referee 2

This paper represents a theoretical study of multiexciton generation (MEG) in model type-II hetero-nanorods (NRs) with both parts comprised of CdSe-like material with identical band gaps (E_g) but an arbitrary band offset (Δ) which is one of the parameters of the calculations. The authors calculate linear absorption spectra of these structures and show that they are virtually independent on Δ . On the other hand, the calculated MEG yields show a considerable increase with increasing Δ , which is interpreted in terms of the effect of increasing electric field. I find these results of potential interest. However, I have several questions regarding their analysis and interpretation.

We thank the referee for his/her supporting assessment of our work.

Specific points of referee 2:

1. *The authors consider that the band gap of their structures is the same as that of CdSe (E_g). However, for type-II hetero-nanocrystals, the actual band gap is defined by the difference in electron and hole energies at the hetero-junction, i.e., $E_{g1,2} = E_g - \Delta$. This energy defines the emission wavelength of hetero-structures as well as the onset of optical absorption. This leads me to the first question: Absorption spectrum in Fig. 1b has a tail extending to 1 eV. It's likely due to the indirect transition at the interface. Why the onset of indirect absorption does not change with Δ ? It must be directly linked to $E_{g1,2} = E_g - \Delta$.*

The referee is asking a very important question. However, before we address his/her point, we would like to correct a minor (but important) observation made: The spectra shown Fig. 1b are plotted versus the energy scales by the bare CdSe gap. Thus, the tail does not extend to 1 eV, but slightly extends below E_g (the same E_g is used for all plots without considering the shift Δ). This tail results for the finite spectral resolution of the calculation, which is based on a real-time stochastic approach, as explained in the “Method” section. If the onset of absorption would have been $E_g - \Delta$, then one would observe a significant shift of onset (rather than a minor tail) since the values of Δ used in the calculation range from $0.15E_g$ to $0.75E_g$. To conclude, we observe no shift in the absorption onset with Δ .

Now, the referee is asking, “why?”. This can be explained by a close examination of the lower panel of Fig. 3 (old right panel of Fig. 2). It is clearly seen that the overlap of the hole state localized on the right material with the electron state localized on the left material is negligibly small. In fact, this is the case for nearly all electron-hole states with energy difference smaller than E_g . Since the overlap is small, the oscillator strength for transition below E_g is small, and the absorption onset does not change significantly. This result also implies that the emission lifetime will be long, but the emission

wavelength will, of course, shift to the lowest energy. For photocurrent conversion, the absorption onset determines the relevant gap while for LED it is the emission, but the latter is not a concern of the current work.

2. *I have a problem with interpretation of the data in terms of electric field. The potential in Fig. 1a indeed resembles the structure of a p-n junction with the internal field within the depletion layer. However, the structures considered in this paper seem to be intrinsic, i.e., do not contain dopants. Hence, there is no field in the structures. Does the Hamiltonian used in the calculations contain the field term? - Probably not. Hence, the field in system is zero. The interpretation in terms of electric field might confuse readers.*

The type of p-n junction we wish to mimic is one that has two different materials (rather than doping the same material with p and n dopants). At the interface of two materials with a different band structure and a different chemical potential, there is going to be an internal field. For reasons that were explained in the paper, instead of using two different materials, we use the same material and **add a field to the Hamiltonian**. The reason is that we wish to study the effect of the internal field on MEG without changing the band structure, which by itself can affect the MEG rates (this has been addressed by us in a recent study, J. Phys. Chem. Lett. 4, 317, 2013). To make this more clear we have modified the second paragraph on page 2, left column, explaining the choice of type-II structure as follows: “To illustrate the core idea and to focus on the effect associated with the role of the internal field, we consider a model system of type-II NR where both building blocks are composed of CdSe with an internal field added to the Hamiltonian at the interface region of width β and maximal value $F \approx \Delta/\beta$, as depicted in the left panel of Fig. 1 (Δ is defined in the caption of Fig. 1). While more realistic models of type-II nanostructures can be handled within the pseudopotential approach, the simplified approach we take allows us to delineate the mechanism of generating multiexcitons induced by the internal field only, without the complexity of the varying band structure and introducing stress and strain across the hetero-junction.”

3. *My impression is that the main reason for observed enhancement is a trivial effect of reduction of the effective band gap ($E_{g_{1,2}} = E_g - \Delta$) when Δ is increased. How do the data in Fig. 2 (left panel) look if they are re-plotted as a function of $E_{g_{1,2}}$? The authors do need to discuss the distinction between the effect of a trivial reduction of the band-gap and some other effects, which might take place in these type-II NRs.*

We would like to note that if the MEG efficiency is scaled by $E_g - \Delta$, which as we argue below, is meaningless, then the onset of MEG would increase with Δ , while the referee suggests that it will not change. Thus, the effect of Δ is rather convoluted and is explained in detail in Fig. 3 and 4 of the revised manuscript and in the discussion around these figures (the discussion starts on page 4, left column, bottom paragraph and ends on page 5, left column, top paragraph).

Scaling the results by $E_g - \Delta$ is wrong since the material is transparent below E_g . It makes sense to scale the results with the optically active gap, as is commonly done for interpreting the role of MEG in nanocrystals (see Beard *et al.*, Nano Letters 10, 3019, (2010)). However, since the optically active

gap is independent of Δ , one can also view the results shown in Fig. 2 on an absolute energy scale (which is also meaningful), and even on an absolute energy scale, the effect of increasing Δ leads to a significant decrease of the onset of MEG. To summarize, increasing Δ decreases the onset of MEG on an absolute energy as well as on an energy scaled by the optical active gap.

4. *The authors briefly discuss the effect of Δ on photovoltage. In fact, I think they have a misprint on p. 4 where they state that the proposed scheme will increase voltage. The proposed scheme actually should decrease the photovoltage in proportion to the introduced band offset Δ . The extracted voltage will be "pinned" by the indirect band gap $E_{g1,2}$. The question is what is the net gain from the proposed scheme in terms of the power output? The authors should discuss this important aspect of their work in the manuscript.*

We agree with the referee regarding the misprint on page 4. Indeed, increasing Δ will decrease the photovoltage. We have corrected this misprint on page 4. Regarding the question on the net gain, this is a difficult one to answer with our current tools, as there are several factors that are unknown and would influence the gain in efficiency due to MEG. For example, one requires a rapid scheme to collect the charge carriers before other recombination processes would take place. Our work, while motivated by the technological implications of MEG, is focused on the basic science. Therefore, at this point, the best we can say regarding the net gain is qualitative and is based on the following principles: (a) Band offsets are required to separate charge carriers in type-II solar cell. (b) Such offsets can become useful in increasing the MEG efficiency if the interface between the two building blocks is sharp and if the exciton cooling rate is slower than 0.1ps^{-1} . Furthermore, to reach the theoretical limit of 45% efficiency will require the design of materials with a gap as small as $\approx 0.7\text{eV}$ (see Beard *et al.*, Nano Letters 10, 3019, (2010)). This is now reflected in the revised discussion at the end of the manuscript (page 5, right column, second paragraph): "Further development is required in order to utilize the current scenario in type-II heterojunction solar cell to improve its efficiency. For an optimal performance, one requires the design of materials with gaps close to 0.7eV , a sharp interface between the building blocks, and a moderate value of the band offset, Δ , to avoid reducing significantly the photocurrent voltage. Moreover, an efficient and rapid scheme to collect the multiple charge carriers is required before Auger recombination and other relaxation process step in. While some of these targets are already achievable by colloidal chemistry synthesis, the reduction of the cooling rate and the rapid collection of multiple charges requires the development of novel schemes along the lines recently proposed for core/shell nanocrystals."

5. *I am not sure that I understand authors' reasoning regarding slowed cooling and its effect on the overall efficiency. They focus on calculations of the effect near the MEG threshold when following impact ionization the carriers reside in band-edge states. They are already "cooled." Do the authors imply that their scheme is based on both MEG and hot-electron-cell ideas? If so, this needs to be discussed in much greater detail when it is discussed in the present version of the manuscript.*

We are sorry for the confusion. The referee is correct that for the doubly excited states (biexcitons), there is no need to introduce a cooling rate, since they are at the band edge and are assumed to live long enough to be collected and definitely longer than the timescale associated with generating this multiexciton state. Indeed, no such cooling rate is introduced into our theory.

So, why is there a cooling rate in the theory? It is well-known that for efficient MEG, the rate of generating more excitons needs to be larger than the non-radiative exciton cooling rate. However, unlike hot-electron-cell ideas where one needs to collect hot carriers and separate the charge before energy is lost, in MEG-based cell, it is sufficient to duplicate the charge on timescale faster than the cooling rate. Since duplicating the charge is much faster than separating charge carriers, MEG-based cell ideas still hold the promise of breaking the SQ limit, while hot-electron-cell ideas have long been abandoned.

To avoid confusion by other readers, we have slightly modified the second paragraph of the introduction on page 1, breaking it into two paragraphs, where the first reads: “The necessary conditions for MEG to be technologically useful are that the onset energy E_{on} be close to the lower threshold of twice the quasiparticle band gap ($2E_g$) and that beyond this onset the MEG efficiency (i.e. the average number of excitons produced by each absorbed photon) increases rapidly with photon energy. In order to push the onset of MEG to $2E_g$, the rate of generating multiexcitons needs to exceed the rate of the competing non-radiative exciton cooling channel. Achieving this goal is very different from older concepts of collecting hot carriers, since the timescales for separating charge carriers are very long compared to the timescales of carrier multiplication by multiexciton generation.”

Referee 3

In the present manuscript, Prof Rabani and his co-workers have provided a theoretical treatment of the pathways to increase the efficiencies of multiple exciton generation (MEG) in quantum nano systems. The authors have considered CdSe nanorod as a model system and found that the MEG efficiency can be increased by a significant amount beyond Shockley-Queisser limit, by considering the joint effects of the internal field and slowing down the exciton cooling rate. They have also rationalized type-II heterostructure as the primary requirement for increasing the efficiency of MEG. The role of the finite band offset in type-II heterostructures to separate the charge carriers and to reduce the electron-phonon couplings for increasing the MEG efficiency has been described in detail. Finally, the authors have shown the variation of both the coulomb couplings as well as the trion density of states as a function of the onset energy to show their effects on MEG efficiency.

The present manuscript deals with a very timely problem in the field of solar cells where increasing the efficiency of the solar cell performance beyond the Shockley-Queisser limit (31%) is one of the most challenging problems to the entire scientific community involved. This work presents a unique idea for increasing the efficiency of MEG in quantum nano systems. The authors provide valuable information regarding the hybrid type-II nanorod and how the MEG efficiencies are related to the internal field, coulomb coupling and density of the trion states. This work will certainly open up a new direction and would be very much interesting to both experimental and theoretical researchers.

We thank the referee for his/her positive assessment of our work.

Specific points of referee 3:

1. *The results are convincing and authors have provided sufficient evidences for drawing the conclusions. The theoretical methods developed by Rabani et al. are already well established. But, it will be more beneficial to the general readers if authors discuss a little bit about the critical limit of the parameters (such as internal field etc.) relating to the efficiencies of MEG near $2E_g$.*

Please see reply to comment 4 by referee 1.

2. *Also it will be very interesting (if possible) to emphasize the trend of MEG efficiencies by varying the size of the nanorod studied.*

This is an excellent point which was recently addressed by us. In short, we were surprised by experimental observations reporting that MEG is more efficient in rods than dots. Due to linear momentum conservation, we had believed that MEG efficiency in rods will be lower than dots. We carried out calculations and surprisingly found that the MEG efficiency is independent of the rod length (but does depend on its diameter). Ref. 39 of our revised manuscript provides the detail and the analysis of this surprising result. Later, an experimental observation reporting on MEG efficiencies in rods confirmed our predictions (Ref. 40 in our revised manuscript).

3. *In figure 2, left panel, both the upper and lower results show the biexciton generation efficiency as a function of energy for different values of Δ and for two different values of β . It will be nice if the number of curves were same for both the cases. In addition, the whole figure needs better representation and clarity.*

First, we note that the set of parameters shown in the lower panel of Fig. 2, the effect of Δ on the MEG efficiency is negligible for values smaller than 0.45. We have not computed the results for $\Delta = 0.15$ and $\Delta = 0.30$ (but we did compute the results for $\Delta = 0$). Each data point takes 2 month on a parallel cluster computer, and thus cannot be carried out at this point. However, we have pointed this out in the caption of the revised Fig. 2 on page 3: “Note that the results for the two lower values of Δ for $\beta = 2.5$ nm are similar to those of $\Delta = 0.45$, and thus are not shown.”

We have also improved the presentation of the figure, as also requested by referee 1 and broke it to two separate figures. The new Fig. 2 contains only the left and middle panels of the old Fig. 2 and the right panels of the old Fig. 2 have moved to a new Fig. 3.

4. *In figure 3, left panel, density of both positive and negative trion states are shown with different Δ values. The negative trion densities for various Δ values follow almost the same trend, whereas positive trion density states vary with different Δ values. Please provide explanation of reasons behind this. Also, the value of density is higher for negative trion than that of the positive one in particular around the onset energy (E_{on}). Please make a brief discussion on this.*

We have clarified in the figure caption (now figure 4, page 4) that the positive (hole) DOTS are plotted on negative energies and the negative (electron) DOTS are plotted on positive energies. We are sorry for not clarifying this in the original version (it was discussed in the main text but not in the figure caption), but this is the convention used in the MEG community (since the holes have energies mea-

sured below the top of the valance band and thus, are negative). This implies that the positive DOTS are higher than the negative DOTS.

Regarding the difference between the positive and negative DOTS, we note that the DOTS is given by a triple convolution of the DOS (shown in the figure above, with the top of the valance band at $\approx -3E_g$). The positive DOTS is given by a triple convolution of the DOS of two holes (DOS below the Fermi energy) and one electron (DOS above the Fermi energy) and the negative DOTS is a triple convolution of the DOS of two electrons and one hole. The differences between the positive and negative DOTS can be traced to the differences between the electron/hole DOS. Specifically, the positive DOTS is higher results from the higher hole DOS at the relevant energy regime. However, we wish to stress here that the important effect of Δ on the DOTS is the increase of the DOTS near $\varepsilon = \pm E_g$, which is clearly observed in the Fig. 4 (note that the DOTS is plotted on a semi-logarithmic scale).

We have added a short discussion on page 5, top paragraph of the left column, to explain the behavior of the DOTS shown in Fig. 4: “...The DOTS is obtained by a triple convolution of the density of states (DOS) not shown here. For the negative DOTS, the triple convolution involves the DOS of two electrons (i.e., the DOS above the Fermi energy) and one hole (i.e., the DOS below the Fermi energy) while for the positive DOTS it involves the triple convolution of the DOS of two holes and one electron. Common to both DOTS is the increase in $\rho_T^\pm(\varepsilon_e)$ at $\varepsilon = \pm E_g$ above/below the conduction/valance band minimum/maximum. Note that we use a semi-logarithmic scale and thus, the magnitude of the effect is rather dramatic with increasing bias. The positive DOTS is typically higher than the negative DOTS at the energies measured from the corresponding band edge, a result that can be explained by the larger DOS of holes. Furthermore, the negative DOTS shows a stronger dependence on Δ resulting from the structure of the DOS of holes versus electrons.”

Reviewer #1 (Remarks to the Author)

The authors have addressed each of the points that I raised in my review, and I now support publication of this manuscript in Nature Communications. Oh, a spurious "D" has been inadvertently prepended to the title.

Reviewer #2 (Remarks to the Author)

I appreciate the revisions and clarifications made by the authors in the newly submitted paper. However, I still have a few comments that should be addressed before I can make a decision on recommending (or not) this paper for publication.

At present, the focus of the paper is on advantages of the proposed MEG scheme; however, its potential deficiencies are not thoroughly discussed. I agree that the introduction of the electric field does reduce the MEG onset and increase the biexciton yield. However, it is not clear that this scheme will lead to the enhanced power conversion efficiency (PCE), the quantity of primary importance in practical PVs. The problem is that the introduction of the type-II offset reduces the open-circuit voltage (V_{oc}) and also opens a "transparency" window right above the indirect band gap. In principle, these two factors can overwhelm any MEG-related boost in the photocurrent and lead to the drop of the power output instead of increasing it. Let me elaborate on these points.

1. In responding to my comment on the effect of Δ on absorption onset, the authors explain that it is defined not by the indirect band gap $E_{g1,2} = E_g - \Delta$, but the "optical gap", E_g . The rationale for this behavior is a weak overlap between near-band-edge electron and hole states. I agree, this is a valid explanation. However, this also suggests that the increase in Δ leads to the increase in the width of the "transparency" window above the indirect band gap, which leads to reduced light harvesting. This point needs to be discussed in the manuscript and analyzed in quantitative terms.

2. The authors should also discuss and analyze the effect of increasing Δ on V_{oc} . In the response to my original comment, they recognize the detrimental effect of Δ on V_{oc} . However, they do not analyze it quantitatively. On the other hand, the authors have all information required for calculating PCE as a function of Δ . They can use the calculated absorption spectra (Fig. 1) and MEG efficiencies (Fig. 2) for deriving the spectra of external quantum efficiency (EQE) and then a short-circuit current (J_{sc}). Using J_{sc} and values of the indirect band gap, they can easily calculate V_{oc} and the fill factor, and finally, the PCE. These calculations will provide an unambiguous answer to the question on whether the proposed scheme is of practical value from the point of view of real-life PVs.

I'll be happy to examine the revised manuscript again after the authors address the above comments.

Reviewer #3 (Remarks to the Author)

The authors have now modified the manuscript according to my previous suggestions. So, I recommend the paper to be published as it is.

In response to Referee 2:

1. *At present, the focus of the paper is on advantages of the proposed MEG scheme; however, its potential deficiencies are not thoroughly discussed. I agree that the introduction of the electric field does reduce the MEG onset and increase the biexciton yield. However, it is not clear that this scheme will lead to the enhanced power conversion efficiency (PCE), the quantity of primary importance in practical PVs. The problem is that the introduction of the type-II offset reduces the open-circuit voltage (V_{oc}) and also opens a "transparency" window right above the indirect band gap. In principle, these two factors can overwhelm any MEG-related boost in the photocurrent and lead to the drop of the power output instead of increasing it. Let me elaborate on these points.*

We first would like to note that in any solar cell device, there are losses due to the open circuit voltage. We are not introducing an open circuit voltage. We are taking advantage of the fact that it exists. And given that it exists, it is certainly clear utilizing MEG will increase the power conversion efficiency of the device. It may result in a negligible effect if the MEG efficiency at twice the gap is small, but may also result in an increase of several percent in the power conversion efficiency if MEG is significant at twice the gap. This is all we claim. We further discuss this point below, when we reply to comment 2 of the referee.

2. *In responding to my comment on the effect of Δ on absorption onset, the authors explain that it is defined not by the indirect band gap $E_{g1,2} = E_g - \Delta$, but the "optical gap", E_g . The rationale for this behavior is a weak overlap between near-band-edge electron and hole states. I agree, this is a valid explanation. However, this also suggests that the increase in Δ leads to the increase in the width of the "transparency" window above the indirect band gap, which leads to reduced light harvesting. This point needs to be discussed in the manuscript and analyzed in quantitative terms.*

We are happy to see that the referee agreed fully with our claim that the two parts of the junction are optically decoupled at frequencies between $E_g - \Delta$ and E_g (where E_g is the material gap and Δ the potential drop across the heterojunction) which he terms the transparency window. We also agree with the referee that increasing Δ leads and an increase of what he or she refers to as the “transparency” window. However, the optical onset of absorption does not change with Δ and thus, the optical properties of our system are Δ -independent, as clearly shown in Fig. 1. We do agree with the referee that increasing Δ will result in a reduced light harvesting, but mainly due to the increase of the open circuit voltage. This is true regardless of the effect of MEG and has been discussed quantitatively by many authors, see for example L. C. Hirst and N. J. Ekins-Daukes, “Fundamental losses in solar cells,” Prog. Photovolt: Res. Appl., vol. 19, pp. 286-293, 2011.

The main point, however, of our manuscript, is that in type-II single junction cells, there is a finite value of Δ which can be utilized to generate multiexcitons and increase the efficiency of the cell for a fixed value of Δ . To emphasize this, we have added on the revised manuscript (on page 4, second column) the following paragraph: “While a finite value of Δ is required to separate the charge carriers in type-II solar cell junctions, the value of Δ used to increase the MEG efficiency as depicted in the left panels of Fig. 2, will lead to a significant decrease in the photovoltage and as a result, to an overall lower solar cell performance [50]. As shown in the right panels Fig. 2, this issue is mitigated when the exciton cooling lifetime is increased above 10ps (which, in fact, can be induced by an pronounced electron-hole charge separation...”

3. *The authors should also discuss and analyze the effect of increasing Δ on V_{oc} . In the response to my original comment, they recognize the detrimental effect of Δ on V_{oc} . However, they do not analyze it quantitatively. On the other hand, the authors have all information required for calculating PCE as a function of Δ . They can use the calculated absorption spectra (Fig. 1) and MEG efficiencies (Fig. 2) for deriving the spectra of external quantum efficiency (EQE) and then a short-circuit current (J_{sc}). Using J_{sc} and values of the indirect band gap, they can easily calculate V_{oc} and the fill factor, and finally, the PCE. These calculations will provide an unambiguous answer to the question on whether the proposed scheme is of practical value from the point of view of real-life PVs.*

The referee is eager to know whether the proposed scheme is of practical value from the point of view of real-life PVs. This is an interesting engineering question, but it is not the focus of our more basic science study. The paper is about discovery of a new physical effect in semiconductor nanostructures, with *potential* implications to solar cells, which will need to be explored in future applications. We have no claims concerning power conversion efficiency (PCE) of a device based on the specific active material we considered. The only claims we made were that (a) the PCE will increase by utilizing MEG and that MEG can be made efficient at twice the gap. The latter was also discussed quantitatively.

We refer the referee to the concluding paragraph regarding the limitations of the current discovery, which we believe provide an honest opinion of the open challenges utilizing the proposed MEG sce-

nario: “Further development is required in order to utilize the current scenario in type-II single cell junctions. For an optimal performance, one requires the design of materials with gaps close to 0.7 eV, a sharp interface between the building blocks, and a moderate value of the band offset, Δ , not to reduce significantly the photocurrent voltage. Moreover, an efficient and rapid scheme to collect multiple charge carriers is requires before Auger recombination and/or other relaxation process step in. While some of these targets are already achievable by colloidal chemistry synthesis, the reduction of the cooling rate and the rapid collection of multiple charges requires the development of novel schemes along the lines recently proposed for core/shell nanocrystals.”